# Formative Assessment of Diagnostic Testing in Family Medicine with Comprehensive MCQ Followed by Certainty-Based Mark

**DOI:** 10.3390/healthcare10081558

**Published:** 2022-08-17

**Authors:** Charles Herbaux, Aurélie Dupré, Wendy Rénier, Ludovic Gabellier, Emmanuel Chazard, Philippe Lambert, Vincent Sobanski, Didier Gosset, Dominique Lacroix, Patrick Truffert

**Affiliations:** 1Clinical Hematologic Department, University of Lille, CHU Lille, ULR 2694 Metrics, F-59000 Lille, France; 2Clinical Hematologic Department, University of Montpellier, CHU Montpellier, F-34000 Montpellier, France; 3Laboratoire CIREL (EA 4354), Service Conseil et Accompagnement à la PEdagogie (DIP-CAPE), University of Lille, F-59000 Lille, France; 4CERIM Public Health Department, University of Lille, CHU Lille, ULR 2694 Metrics, F-59000 Lille, France; 5General Medicine Department, University Montpellier, CHU Montpellier, F-34000 Montpellier, France; 6Internal Medicine Department, University of Lille, CHU Lille, ULR 2694 Metrics, F-59000 Lille, France; 7Lille University School of Medicine, University of Lille, CHU Lille, ULR 2694 Metrics, F-59000 Lille, France; 8Pediatric Department, University of Lille, CHU Lille, ULR 2694 Metrics, F-59000 Lille, France

**Keywords:** education, diagnostics

## Abstract

Introduction: The choice of diagnostic tests in front of a given clinical case is a major part of medical reasoning. Failure to prescribe the right test can lead to serious diagnostic errors. Furthermore, unnecessary medical tests are a waste of money and could possibly generate injuries to patients, especially in family medicine. Methods: In an effort to improve the training of our students to the choice of laboratory and imaging studies, we implemented a specific multiple-choice questions (MCQ), called comprehensive MCQ (cMCQ), with a fixed and high number of options matching various basic medical tests, followed by a certainty-based mark (CBM). This tool was used in the assessment of diagnostic test choice in various clinical cases of general practice in 456 sixth-year medical students. Results: The scores were significantly correlated with the traditional exams (standard MCQ), with matched themes. The proportion of “cMCQ/CBM score” variance explained by “standard MCQ score” was 21.3%. The cMCQ placed students in a situation closer to practice reality than standard MCQ. In addition to its usefulness as an assessment tool, those tests had a formative value and allowed students to work on their ability to measure their doubt/certainty in order to develop a reflexive approach, required for their future professional practice. Conclusion: cMCQ followed by CBM is a feasible and reliable evaluation method for the assessment of diagnostic testing.

## 1. Introduction

The choice of diagnostic tests in front of a given clinical case is a major part of medical reasoning. Failure to prescribe the right test can lead to serious diagnosis delays and errors, especially in family medicine. Furthermore, unnecessary medical tests are a waste of money and could possibly generate injuries to patients [1,2]. Using open questions or modified essay questions [3] to evaluate the choice of laboratory and imaging studies allows for an unlimited range of answers but is time consuming to mark and present considerable variation in standards of marking [4]. New methods can consistently assess clinical reasoning, for example, with think-aloud and concept-mapping protocols [5]. However, given their complexity, these evaluations are difficult to apply to the regular assessment of large numbers of students. Thus, multiple-choice questions (MCQs) are frequently used during university tests and commonly in medical studies examinations in France. They often require less time to administer for a given amount of material than would tests requiring written responses and they have a high level of reliability. Moreover, they display high potential time–economy advantages for the correctors. In contrast, MCQs allow sight recognition or random guessing [6,7,8]. The majority of MCQs test lower-order thinking skills (recall and comprehension) rather than higher-order skills, such as application and analysis. Nevertheless, it has been shown that a well-structured MCQ has the ability to assess higher-ordered thinking [9], as described by Bloom’s taxonomy [10,11]. Creating high-quality MCQ is time consuming and the work of MCQ authors needs to be constantly and carefully assessed [12,13,14].

Certainty-based marking (CBM) is another tool that can be used in the educational field. CBM is also known as confidence-based marking. A question is added for each answer, asking the student to assess the level of confidence in the given answer (i.e., “are you sure at 60%, 80% or 100%?”) [15,16]. This technique has been used in medical education to encourage reflection on reasoning prior to making clinical-based decisions [17,18] and it has already been associated with MCQs [19]. It is thought to enhance deeper levels of learning at the expense of common learning practices [20,21].

In an effort to improve our students’ training to diagnostic testing, we implemented a specific type of MCQ, with a high number (107) of options matching various basic medical tests. We called this assessment method “comprehensive” MCQ (cMCQ), because all possible options reasonably needed by a general practitioner for a given outpatient situation are given. cMCQ was then followed by a CBM. The aim of the present study was to explore the feasibility and reliability of cMCQ/CBM to assess the choice of laboratory and imaging studies in front of various clinical cases.

## 2. Methods

### 2.1. Participants

This prospective study was conducted in October 2017, at Lille University School of Medicine. The study was approved by the local Institutional Review Board (“*Conseil de Faculté*”). Students were informed orally and with a written protocol about study proceedings, on a voluntary basis. cMCQ/CBM scores did not account for official faculty assessment. The experimental questions were submitted to students right before the standard tests. All data were anonymously analyzed. Overall, out of the 498 students attending their sixth year of medical school, 456 students agreed to participate (91%) and were included in this study. In the whole year group, twenty-three students were not attending the faculty test and nineteen denied participation. The results of this study were presented to students at the end of the academic year.

### 2.2. Procedure and Data Collection Method

Each question in the experimental assessing method was composed of three different parts. The first one was a short clinical scenario, ending with one of the two following questions: “In this situation, which diagnostic test(s) is (are) indicated?” or “What are the “x” diagnostic tests that can confirm your main hypothesis?”, where “x” was a number set by the author of the cMCQ. The second part was the list of possible diagnostic tests to prescribe (Appendix A). This list included 108 proposals, corresponding to 107 basic medical tests, as well as the following proposal “No further examination is necessary”. They were written to cover all possible options reasonably needed in front of a consultation in family medicine. The third part consisted of a request for degrees of certainty regarding the previous answer (CBM). The student could choose one out of six different degrees of certainty allocated to their response (i.e., 0%, 20%, 40%, 60%, 80% and 100%). This numerical method is described to be more relevant in the CBM [21]. From a teaching point of view, asking students to assess their degree of confidence allows us to distinguish 4 types of answers [22]: a/ “serious mistake”: error made with high degree of certainty; b/“mastered answer”: correct answer with high degree of certainty; c/ “ignorance”: error made with low degree of certainty; d/“weak knowledge”: correct answer given with low degree of certainty. Responses falling into the two last categories are strongly influenced by randomness.

Only the first part of the cMCQ was modified from question to question, parts 2 and 3 remained unchanged. Two examples of cMCQ/CBM are presented in Appendix B. The terms and conditions of this experimental assessing method were explained to students one month before the tests. At that moment, they also received the list of laboratory and imaging tests (Appendix A), which allowed them to familiarize themselves with the list in order not to waste time discovering all 107 proposals. Three minutes were allowed for each question. Two sessions of fifteen minutes each (five cMCQ/CBM) were submitted to students, right before two standard faculty assessments.

### 2.3. Scoring Scale

The scoring scale of cMCQ depended on the number of correct answers and was empirically chosen. In case of a single correct answer, if this answer was chosen, 1 point was awarded. Knowing that the score could not be negative, 0.2 points were removed for any incorrect answer ticked. In case of multiple correct answers, 1 point divided by the number of expected answers was awarded for each good answer (for example, when there were 4 correct answers, each one valued 0.25 points). Knowing that the score could not be negative, 0.05 points were removed for any incorrect answer ticked. The score for the total of 10 cMCQ was converted out of 100. In addition to this main mark, the CBM could lead to a maximum of 2 “bonus” points. The average of all the certainty levels for *correct* answers was calculated and corresponded to a maximum of 1 bonus point. For example, if the average of all the certainty levels for correct answers was 90%, the bonus was 0.9 points (90% = 0.9). Similarly, the average of all the certainty levels for *incorrect* answers was calculated and subtracted from 1, which corresponded to a maximum of 1 bonus point. For example, if the average of all the certainty levels for incorrect answers was 30%, the bonus was 0.7 points (30% = 0.3; 1 − 0.3 = 0.7). Of course, this scoring scale values the right answers with high degrees of certainty (“mastered answer”). Nonetheless, and this is more unusual and, therefore, one of the originalities of this work, the wrong answers given with low degrees of certainty (“ignorance”) were also valued. It is assumed that in a real professional situation, the student would have tried to gather additional information or asked for a peer’s opinion. This scale was designed to minimize the influence of the assessment of CBM on the student’s final score. It mainly aimed to enlarge score distribution and to separate students with equal or very close scores.

### 2.4. Faculty Tests and Context

Standard exams consisted of three-hour sessions with 120 MCQs in the format imposed for the ECN (*Épreuves Classantes Nationales*), the national medical examination which takes place at the end of the sixth year of medical school in France. Each MCQ included a description of a short clinical scenario followed by five sentences. One to four of these sentences were correct and one to four were incorrect. Two examples of these MCQs, which will be called “standard MCQ”, are presented in Appendix C. The scoring scale was as follows: no mistake: 1 point; 1 mistake: 0.5 points; 2 mistakes: 0.2 points; 3 mistakes or more: 0 points. Both tests, experimental and standard, were conducted on a tablet computer (iPad^®^, Apple Inc., Victor Valley, CA, USA). According to the rules of the ECN, the evaluation by MCQ on tablet computer is mandatory in France. This allowed us to use those tablet computers for the cMCQ and CBM. The themes of cMCQ were identical to those of the faculty tests (rheumatology and infectious diseases).

### 2.5. Statistical Analysis

We first performed a descriptive analysis on variables of interest. Qualitative variables were quoted as the frequency (percentage). Quantitative variables were quoted as mean ± standard deviation (SD) when normally distributed or as the median (interquartile range (IQR)) otherwise. The 95% confidence interval (CI) was calculated using a normal distribution. Independence between categorical variables was tested using a chi-squared test or Fisher’s exact test. Independence between categorical and quantitative variables was tested using Student’s *t*-test or analysis of variance. Independence between quantitative variables was tested using the Pearson correlation coefficient nullity test. Their relationship was modeled using linear regression. There were no missing data. All tests were two-sided and the threshold for statistical significance was set to *p* < 0.05. Analyses were performed using R software (version 3.3.1, R Foundation for Statistical Computing, Vienna, Austria).

## 3. Results

The mean age of the 456 students was 24.5 (±2.0) years. During the first session of 15 min, the average time spent on the five experimental questions was 9 min 26 s (±2 min 30 s), with a minimum of 4 min 6 s and a maximum of 15 min. Through the second session, the average time was 8 min 27 s (±2 min 23 s), with a minimum of 3 min 2 s and a maximum of 15 min. The mean score of the two cMCQ/CBM sessions was 57.7 points (±13.1); the distribution of the scores is presented in Figure 1A. In comparison, mean score of the two standard MCQ sessions was 58.9 points (±7.3); the distribution of the scores is presented in Figure 1B.

Figure 2 shows a scatter plot and regression line of experimental and standard scores. We found a significant correlation between these two parameters (*p* < 0.0001) and the proportion of “cMCQ/CBM score” variance explained by “standard MCQ score” was 21.3% (r^2^ = 0.213).

Regarding the CBM, the degrees of certainty were significantly higher for correct answers (median and quartiles: 90% (80%; 100%)) vs. incorrect answers (70% (60%; 80%)) (*p* < 0.0001; Figure 3). “Serious mistakes” were easily identified, corresponding to false answers with high level of confidence. For example, 5.2% of the whole cohort answered the first question incorrectly with a certainty level of 100%.

## 4. Discussion

We described the usage of a specific type of MCQ with a high number of options matching a comprehensive list of diagnostic tests, followed by a certainty-based marking. This assessment tool was feasible in sixth-year medical students to assess the choice of laboratory and imaging studies in front of a clinical case of general practice. Our prospective study was performed on a large number of students, which allowed us to show that cMCQ and CBM can be easily implemented in a medical school using MCQs on a regular basis. As the first part of the cMCQ (the short clinical scenario) is the only one to change from question to question, the framing of a well-structured cMCQ is easy to learn and to perform for teachers. Given the large number of proposed responses, sight recognition or random guessing were minimized [23]. Finally, thanks to the CBM, “serious mistake”, “mastered answer”, “ignorance” and “weak knowledge” could be identified among the students.

Score distribution was larger with experimental evaluation than with standard faculty exam. This would facilitate student ranking by reducing the rate of close or identical scores. Furthermore, cMCQs are very similar to diagnostic test choices in real medical practice, since most of test prescription is now computer based, offering similar lists of laboratory and imaging studies. We observed a significant correlation between the results of standard and experimental methods, presumably because of the impact of the skills and knowledge of students to correctly answer both MCQ types. Themes in both assessment methods were matched to minimize the variability that could have been brought by unequal theme knowledge. The proportion of “cMCQ/CBM score” variance explained by “standard MCQ score” was 21.3%. The proportion of unexplained variance stayed, therefore, relatively high, showing that these two methods are different and do not explore the same knowledge. This can also be explained by differences in skills that are assessed by the two methods. In addition to its usefulness as an examination, the results of cMCQ/CBM can be used for other teaching purposes. Mainly, “serious mistakes” feedback could be recognized individually, in order to deconstruct false knowledge. Of course, a high frequency of 100% certainty level for an incorrect answer may only indicate a “serious mistake” once a poorly asked question or erroneous correction was discarded. More broadly, the CBM can also help in developing a reflexive attitude, taking doubt into account, which is essential in medical practice [24]. In that regard, the CBM exercise—by valuing “ignorance” in the scoring scale—induces another relationship to error and can, therefore, facilitate the right attitude to adopt in real practice: gathering literature or seeking advice from a peer. On the other hand, being sure of a correct answer indicates potentially lasting knowledge, which is also valued in our scoring scale. We were surprised by the high overall levels of certainty. This could be explained by insufficient explanations about CBM from our part to the students. Furthermore, this was the first time students were confronted with this CBM exercise. Students could have been overconfident, which is not uncommon among young doctors. Finally, the thorough preparation of medical students for this major testing can also be an explanation, as this sixth-year exam is the most important in France, basically orienting the rest of the students’ professional lives by the ECN final national ranking [25].

Our study has some limitations. Firstly, there is not a specific tool that is fundamentally objective to assess whether one evaluation method is better than another. Some studies claim that the optimum number of options for an MCQ is three [26,27], while our study used more. However, these conclusions should not be of concern, because those studies assessed the single best response in MCQ, with two distractors and one correct answer. It seems reductive to limit the problematic nature of the choice of paraclinical exams to one correct answer to pick up in three propositions. Secondly, as mentioned above, our scoring system was empirically chosen. The number of points withdrawn per false proposal selected could have been different, as well as the number of bonus points for the CBM or the level of certainty to identify serious mistakes. Other scales could have been chosen to give greater weight to CBM [28]. However, it can be easily adapted for faculty assessment goals. Then, the optional nature of this experimental evaluation may have influenced the results but including an experimental evaluation in the official faculty evaluation did not seem appropriate to us. Finally, the results of the CBM would certainly have been different if the students had been able to practice this new exercise, which is difficult to apprehend given the usual university context, where the goal is always to “find the right answer”. From a purely teaching point of view, it would have also been interesting to ask the students about their experience during the exam. Future studies will include such an assessment, such as: “Were you unsettled by the large number of proposals?”; “Were you unsettled by the CBM?”; and “Did you find this mode of evaluation relevant?”.

To better understand the high overall levels of certainty, students from other fields should also be evaluated with similar methods. Thus, as future scientists or biology teachers, the levels of confidence should vary and perhaps subgroups would be identified, based on the type of careers aimed at by students. Indeed, we expect to uncover significant discrepancies and probably even hallmarks of the different university subjects when testing other students of other series or even medical students undertaking other examinations. The results we obtained could be, in part, explained by self-confidence. Hence, marked differences, indeed, appeared in regards to self-esteem between students of health sciences in another country [29]. Due to the impact of the ranking in “ECN” tests on French medical students’ future lives and careers, other students could also be less prepared when taking their own examinations. Moreover, medical doctors are worldwide supposed to be self-assured, so as to inspire trust in their diagnoses and prescribed treatments [30]. Even if self-esteem seems to be impacted by gender, being significantly lower in female students [31], the ability of the future medical doctors to convince their future patients remains essential. However, as clinical decisions require insight and foresight, when both are lacking, overconfidence and error might appear [19]. When self-assurance can be beneficial, overconfidence might also be problematic. 

Overall, we believe that this work is of interest in three ways. First, the “comprehensive” MCQ placed students in authentic situations, close to professional life. Then, CBM brought an important dimension to the test and can also be a learning tool. Indeed, doubt management is a skill that is particularly necessary in the field of medicine, where students (and doctors) must constantly update their knowledge. Finally, this type of examination test is easy to set up and retains the advantages of the MCQ format. The use of this relatively innovative method was found to be a feasible and reliable tool to assess the choice of laboratory and imaging studies in family medicine. Nevertheless, this method of evaluation, using a comprehensive list of choices followed by a CBM, can be easily applied to other fields of medicine. Here, we chose 107 general medical tests, tailored to the practice of general practice, which is thought to be appropriate to sixth-year medical students in France. Hence, we can definitely create a list of tests adapted to each medical specialty (hematology, neurology, etc.). Furthermore, such a methodology could also be applied to the choice of clinical examinations to perform; diagnosis to state; or therapies to propose.

## Figures and Tables

**Figure 1 healthcare-10-01558-f001:**
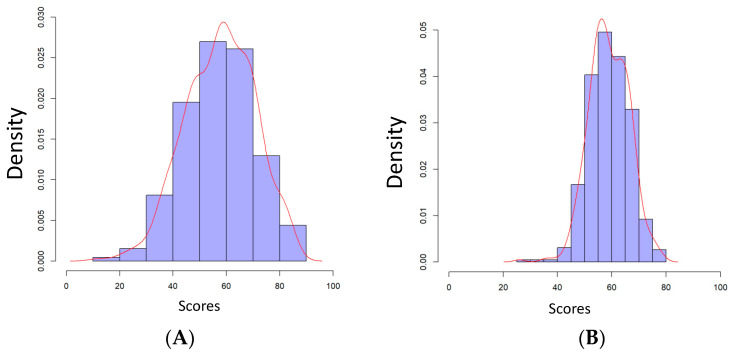
Distribution of the scores obtained with cMCQ/CBM (**A**) and with standard MCQ (**B**). Scores are more spread out with cMCQ/CBM, with a standard deviation of 13.1 versus 7.3 for standard MCQ.

**Figure 2 healthcare-10-01558-f002:**
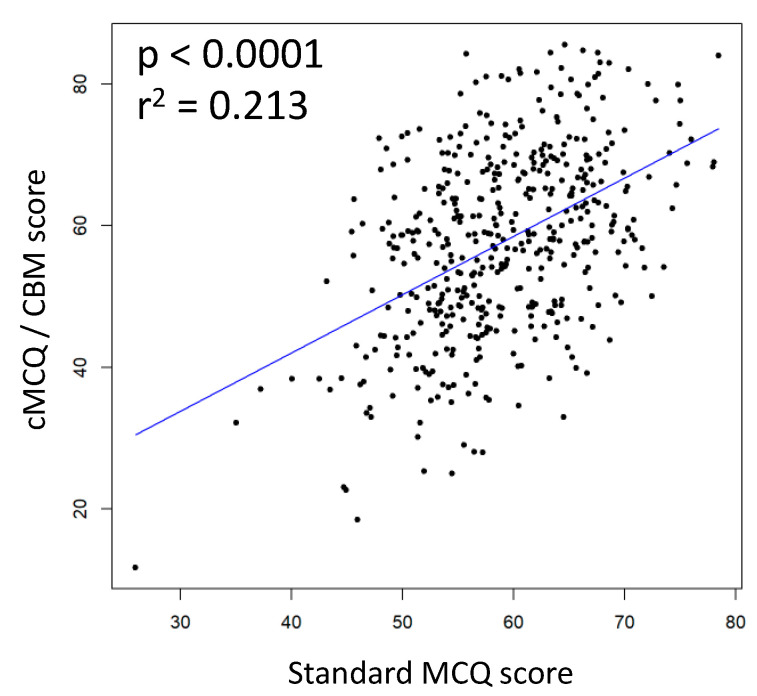
Scatter plot with regression line of individual scores obtained with cMCQ/CBM vs. standard MCQ. R-squared value is denoted by r^2^. The two scores are significantly correlated.

**Figure 3 healthcare-10-01558-f003:**
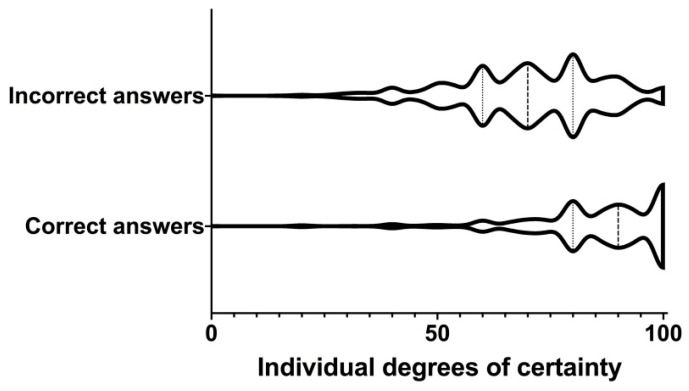
Violin plots of individual levels of certainty for correct and incorrect answers. The dashed lines show the medians and the dotted lines the lower and upper quartiles. Although the levels of confidence appear high for incorrect answers, they are significantly higher for correct answers (*p* < 0.0001).

## Data Availability

Raw data supporting reported results can be requested from the corresponding author by email at c-herbaux@chu-montpellier.fr.

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
