# Peer review of "Formative Assessment of Diagnostic Testing in Family Medicine with Comprehensive MCQ Followed by Certainty-Based Mark"

_healthcare, 2022, doi:10.3390/healthcare10081558_

Round 1
Reviewer 1 Report
1. If you wish to publish in English, please have a native speaker edit - many awkward phrases that detract from comprehension.
2. Major substantive issue, is what advantage this shows for use of cMCQ when it closely tracks the standard MCQ. Why bother changing when scores on both are similar? There should be an external standard with which the cMCQ shows better correlation. Perhaps class rank or faculty assessments or performance on a later advanced exam?
3. If students were told in advance these questions would not count, perhaps they did not try as heard as they would otherwise have done. How was this addressed?
Author Response
On behalf of my co-authors, I would like to thank you for the detailed review of our manuscript for consideration of publication in Healthcare Journal. We have revised the manuscript providing clarifications and corrections where necessary. A point-by-point response to all comments is included below. A clean version of the manuscript, as well as one with tracked changes underlined, is provided.
- We appreciate this comment about the importance of language quality in publications. One of the authors, Miss Wendy Renier, is a native English speaker and she has largely participated in the writing of our manuscript. Following reviewer comment, we have revised some formulations, mainly in the results section (see manuscript with tracked changes underlined). If the reviewer wishes to point out other sentences that could be improved, we will be happy to modify them as well.
- We would like to thank the reviewer for these interesting comments and questions. The external standard is the “standard MCQ” used for the official evaluation, described in the “Faculty tests and context” section (Methods) which correspond to “faculty assessment” mentioned in the reviewer suggestions. The proportion of “cMCQ/CBM score” variance explained by “standard exam score” was 21.3%. The proportion of unexplained variance stayed therefore relatively high (78.7%), showing that these two methods are different and do not explore the same skills. We have clarified these points in the discussion (see manuscript with tracked changes underlined).
- Thank you for raising this important point. First, we would like to specify that including an experimental evaluation in the formal faculty evaluation did not seem appropriate to us. It is the purpose of this study to assess the feasibility of the “cMCQ/CBM” before considering their inclusion in an official examination. However, we agree that the optional nature of this experimental evaluation may have influenced the results. We have therefore acknowledged and commented this limitation in the discussion (see manuscript with tracked changes underlined).
Reviewer 2 Report
The manuscript presented is of excellent quality. The theme is of academic and clinical relevance. The theoretical basis is brief and sufficient. The method and analyses well described, allowing the replication of the study by other groups. The results are relevant to the study proposal. There is reflection on the findings and limitations of the study. Congratulations to the authors.
Author Response
We would like to thank the reviewer for this very positive feedback.